# Using an Online Tool to Apply a Person-Centred Approach in Audiological Rehabilitation: A Pilot Study

**Nerina Scarinci** [1,*] , **Kristen Tulloch** [1,2], **Carly Meyer** [1] , **Katie Ekberg** [1] **and Christopher Lind** [3]

1   School of Health and Rehabilitation Sciences, The University of Queensland, St. Lucia, QLD 4072, Australia
2   School of Health and Behavioural Sciences, University of the Sunshine Coast, Moreton Bay,
    Petrie, QLD 4556, Australia
3   College of Nursing and Health Sciences, Flinders University, Adelaide, SA 5001, Australia
*   Correspondence: n.scarinci@uq.edu.au; Tel.: +61-7-3365-3097

**Abstract:** This study aimed to explore the experiences of adult clients with hearing loss and their audiologists in using an online tool, the Living Well Tool (LWT), during initial audiology appointments. The LWT is designed to help clients identify when and where it is most important for them to communicate effectively and live well with hearing loss. A total of 24 adult clients with hearing loss and two audiologists participated in this study. Clients were invited to complete the LWT prior to their next audiology appointment, however, most clients chose to use the LWT in-session with their audiologist. Following the appointment, clients and audiologists participated in individual qualitative semi-structured interviews to explore their experiences of using the LWT, and the extent to which the LWT facilitated person-centred care. Qualitative analysis five key themes which reflected participants' experiences and perceptions of using the LWT: (1) the LWT enhances audiological care; (2) the LWT supports person-centred audiological care; (3) the use of the LWT should be individualised; (4) users value comprehensiveness; and (5) users value accessibility. This study demonstrated that the LWT supported the provision of person-centred audiological care, providing a flexible, comprehensive and accessible means for audiologists to gain an understanding of their clients' needs and preferences. However, it was also noted that the use of a tool must be individualised and accessible for all.

**Keywords:** communication; hearing loss; family-centred care; person-centred care; eHealth; psychosocial

## 1. Introduction

Hearing loss occurs frequently in older adults and has numerous negative consequences, including reduced access to everyday spoken communication, social isolation, depression, poor mental health, and reduced quality of life [1]. Due to the chronic nature of hearing loss, it has long been recognised that audiological rehabilitation should embrace a person-centred approach and focus on the communication needs of clients and how they can live well with hearing loss, rather than solely focusing on the hearing impairment. Person-centred care purports high quality, holistic care where the patient is encouraged to be an active participant in their healthcare, differing from traditional, biomedical models of healthcare where the practitioner was considered the expert [2]. Person-centred care is respectful and responsive to the needs and individual values of clients [3], with the key principles of person-centred care including the reciprocal sharing of information between clients, significant others, and clinicians; patient involvement through active participation and engagement in decision-making; and emotional and physical support through listening and responding to psychosocial concerns [4].

Despite widespread acknowledgement of the importance of person-centred care in audiological rehabilitation, research suggests that hearing healthcare continues to have a biomedical focus. In a video observation study, clients and their significant others often

raised psychosocial concerns related to communication difficulties during appointments, however these concerns were rarely addressed, with audiologists having a tendency to focus on providing technical information and progressing a discussion about hearing aids [5]. Although hearing aids address the sensory consequences of hearing loss by improving audibility, they do not address the full range of communication difficulties experienced by the individual with hearing loss and their communication partners, and thus patients and their communication partners often require additional support, beyond hearing aid fitting, to learn to live well with their hearing loss.

eHealth can be used as a way of enhancing person-centred care in healthcare [6]. eHealth is defined as "the cost-effective and secure use of ICT [information and communication technology] in support of health and health-related fields" [7]. eHealth is increasingly being used with clients and families to encourage self-management and self-directed learning in healthcare, including the use of interactive websites and apps [8]. The benefits of eHealth in hearing healthcare have included improved patient outcomes, care access, and patient satisfaction [9–11], reduced latency to help-seeking [12], reduced hearing disability and increased psychosocial wellbeing [13], and improved patient knowledge, self-efficacy, and skills [10,14]. There has been increased interest in technology and internet-based platforms that primarily address education, information, and hearing rehabilitation [11], including those aimed at increasing the uptake of person- and family-centred care [6,15]. Further, access to a combination of online/eHealth and in-person care in the older population has been shown to improve psychosocial well-being [16].

The adoption of technology by older adults in managing their healthcare is heavily influenced by the platform's perceived usefulness [17], accessibility and capability [18]. Importantly however, research indicates that older adults are receptive to the use of eHealth in hearing care, with a recent study utilising digital tools through a virtual audiology clinic, reporting strong engagement from older users [12]. Given the increasing interest in using eHealth in audiological rehabilitation, Paglialonga and colleagues [11] concluded in a recent review that further research is needed to determine the efficacy of eHealth for older adults in clinical practice.

To support a person-centred eHealth approach to audiological rehabilitation, the Ida Institute, an independent non-profit organisation in Denmark, has developed a range of clinical tools to empower hearing care professionals to engage people with hearing loss and their significant others using a person-centred approach to audiological rehabilitation, and in doing so, better address each client's individual needs [15]. One such tool is the online "Living Well Tool" (LWT) [19]. The LWT can be readily accessed online through the Ida Institute website (https://apps.idainstitute.com/apps/lw_en) (accessed on 1 July 2021). The LWT is designed to promote communication and living well with hearing loss for clients, using an active problem-solving approach to focus on relevant and important communication situations and the biopsychosocial needs of the client, rather than focusing solely on hearing aids and the hearing loss. The online LWT has been designed to empower clients to prepare for the rehabilitation process. This is achieved by encouraging them to think about their communication needs prior to their audiology appointment, with a series of photographs and open-ended questions helping them to identify when and where it is most important for them to communicate effectively and live well with hearing loss. Armed with this information, the client and their audiologist can work together to map out strategies for managing the hearing loss and communication goals in a person-centred manner to ultimately improve communication outcomes [15]. While this tool is used clinically as a means of promoting client involvement in the hearing rehabilitation process, there have been no studies to date examining client and clinician experiences in using the tool, and limited research exploring the impact of being prepared before an initial audiology appointment. Therefore, the current study aimed to explore the experiences of clients with hearing loss and their audiologists in using the LWT tool in the rehabilitation process; and the extent to which the LWT facilitated person-centred care.

## 2. Materials and Methods

### 2.1. Participants

Clients. A total of 24 clients with hearing loss were recruited from two audiology clinics in Brisbane, Australia, through two clinical audiologists who agreed to participate in the study. Inclusion criteria for clients included: (a) aged over 18 years; (b) a diagnosed hearing loss, and (c) functional English proficiency to participate in a qualitative interview. Clinic-based convenience sampling was used to recruit participants, whereby the participating clinicians invited all eligible clients with hearing loss to participate. Clients were also invited by their audiologist to bring a family member to attend the audiology appointment, which two clients chose to do. Of the 24 participants with hearing loss, two participants did not provide full demographic data, thus only the available demographic data are reported here. Of these 22 clients who provided full demographic data, 17 were men and 7 were women, aged between 45–84 years ($M = 66.86$ years, $SD = 11.60$). The majority of participants lived with a spouse/partner ($n = 20$), with only 3 clients reporting that they lived alone. Twelve of the 24 participants reported that they currently wore hearing aids, with appointment outcomes including the provision of: hearing aids ($n = 5$), other hearing devices ($n = 16$), communication program ($n = 3$), and communication strategies ($n = 1$).

Clinicians. The two female clinicians were both experienced audiologists (Clinician 1:18 years' experience; Clinician 2:9 years' experience). Both worked in a metropolitan area; one in a private practice, and the other as a clinical educator at a university clinic.

### 2.2. Materials

The Ida Institute's LWT is an online tool comprising of 11 photographs depicting a variety of communication situations (e.g., a noisy environment, watching television, listening in church, participating in a meeting). Participants were asked to select a photograph depicting a communication situation they felt was most relevant, or upload an image of their own, then type a description of this communication situation and rate how difficult it was for them to communicate in this situation on a 3-point easy/medium/hard scale. Participants were then asked to select the most useful strategy to help manage their hearing loss and resultant communication difficulties in the situation from a list of eight strategies (e.g., positioning, asking for clarification, acknowledge hearing loss) or write their own strategy; and to nominate a person who will help them in the situation (e.g., friend, audiologist, spouse, other). Participants could select as many communication situations as they wanted. See Figure 1 for a sample screenshot of the LWT.

### 2.3. Procedure

Ethical approval for this study was obtained from The University of Queensland Human Research Ethics Committee (approval number 2016001740). Prior to data collection, two authors (KT and KE) conducted a training session with the audiologists regarding how to use the LWT. Data were collected between May and November 2017, with data collection ceasing once intended numbers were reached. There were no restrictions on appointment types, and in most cases, clients were sent an information package prior to their audiological appointment, depending on the preference of the client at the time of recruitment. The information package included an introductory letter, information and consent forms, and the website address for the online LWT. Clients were invited to complete the LWT with their family member prior to their appointment, and to email the results of the tool to their clinician for discussion. Participants who booked their audiology appointment at too short notice for a postal package to be received in time were verbally invited to bring a family member and were provided the information and consent forms on the day of their appointment, prior to its commencement.

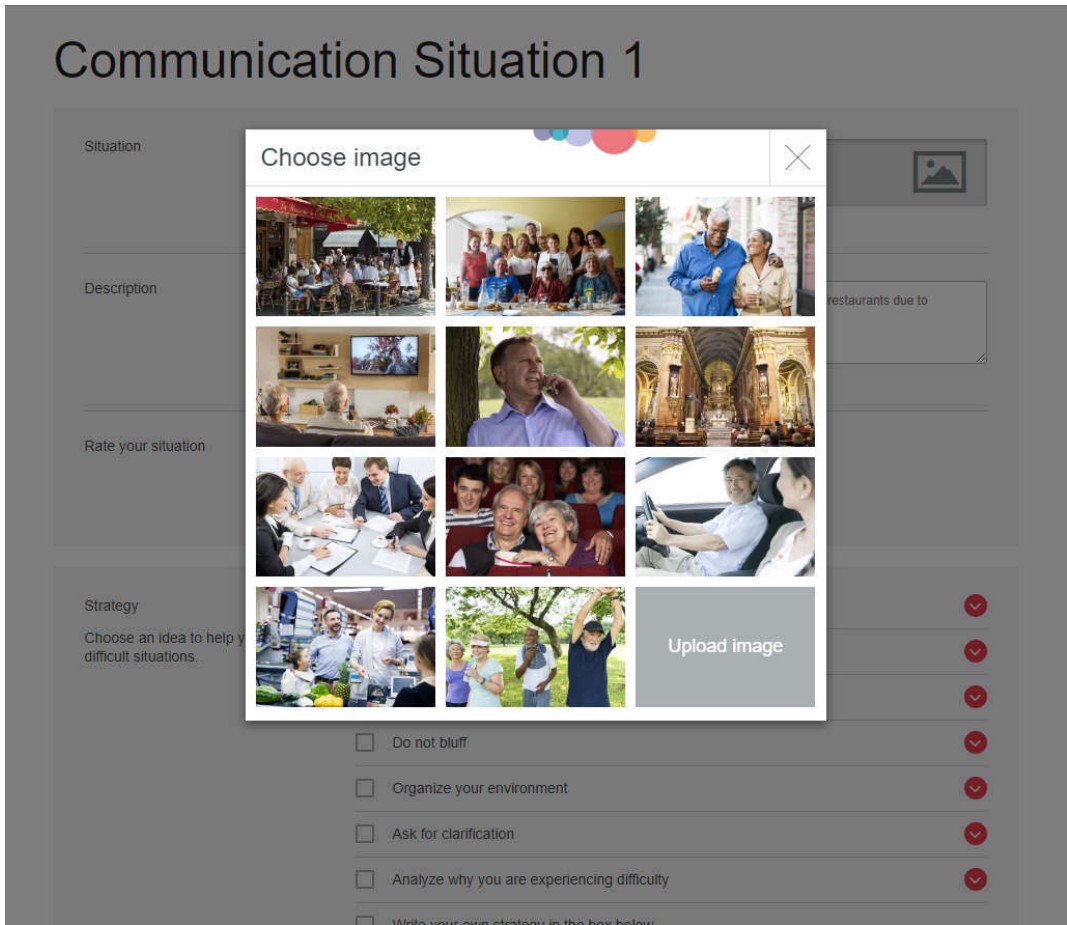

**Figure 1.** Sample screen shot of the online LWT.

A total of 4 out of the 24 clients (16.7%) completed the LWT at home prior to their appointment. Consequently, most appointments (83.3%) commenced with clients completing the LWT in the company of, and sometimes with the assistance of, their clinician. Following their appointment, clients were asked to complete a brief demographics questionnaire. Clients then participated in a semi-structured, qualitative interview about their experiences of using the LWT, either immediately at the clinic or within seven days of their appointment in their own home, pending availability. A topic guide was used to structure the interviews (see Appendix A), with interviews audio-recorded and transcribed verbatim for analysis. The average duration of client interviews was 13 min and 30 s. The two clinicians also participated in a qualitative interview following completion of the project to explore their experiences using the LWT. Each clinician interview lasted approximately 17 min.

*2.4. Data Analysis*

Interviews were analysed using thematic analysis according to the procedures described by [20]. All transcripts were initially read and highlighted to capture an overall impression of the data. Meaning units from the interview transcripts were systematically coded and compiled (KT), followed by a review process of consensus coding (NS, KT) to ensure agreement between authors. An initial grouping of codes into sub-themes and themes was proposed (KT). Collaborative reviews and discussions in regular research team meetings ensured that the themes were continually refined and critically analysed to ensure best representation of the data set.

## 3. Results

Analysis of the interview transcripts revealed five main themes of importance which reflected participants' experiences and perceptions of using the LWT in adult audiological rehabilitation: (1) the LWT enhances audiological care; (2) the LWT supports person-centred audiological care; (3) the use of the LWT should be individualised; (4) users value comprehensiveness; and (5) users value accessibility. Each of these themes and sub-themes is summarized in Table 1 below.

**Table 1.** Summary of themes and sub-themes 1: The LWT enhances audiological care.

| Themes | Sub-Themes |
|---|---|
| 1. The LWT enhances audiological care | LWT complements traditional practice |
| | LWT enhances preparedness for the client and clinician<br>LWT improves appointment efficiency<br>LWT aids communication processes |
| 2. The LWT supports person-centred audiological care | LWT facilitates the client-clinician relationship |
| | LWT makes discussing sensitive matters easier<br>LWT can help clients identify the biopsychosocial impacts of their hearing disability<br>The LWT ensures clients are the priority<br>Relatability of photographs can normalise hearing experiences and help clients identify strategies |
| 3: The use of the LWT should be individualised | The LWT is used in varied ways |
| | Family member involvement was varied<br>Audiologists provide valuable guidance when using the LWT<br>Some LWT users value in-person services |
| 4. Users value comprehensiveness | The LWT is comprehensive |
| | There is potential for the LWT to ask more<br>There is potential for the LWT to give more |
| 5. Users value accessibility of the LWT | Usability of the LWT is valued |
| | The aesthetics of the LWT enhance its accessibility<br>Technology can be challenging |

Theme 1: The LWT Enhances Audiological Care.

The LWT was reported by participants to complement traditional information-gathering practices, facilitate the client-clinician relationship, enhance preparedness for the client and clinician, improve appointment efficiency, and aid communication processes. It could thus be seen as enhancing audiological care.

The LWT complements traditional information-gathering practices. Within the context of typical audiology appointments, participants noted that the LWT was useful in setting the agenda and guiding the case history: *"I think that's [LWT] an improvement on the way it [case history] was done before"* (Client 14), even providing additional case history questions: *"Sometimes you [audiologists] don't always remember to ask all the questions even if you have a sort of template that's being followed, so it's [LWT] probably good"* (Client 16).

Participants also described how the LWT was used to frame discussions around various communication situations, communication strategies, and technological solutions for improving communication in these situations. Clients and clinicians alike noted the usefulness of the LWT in helping clients to identify potential communication strategies: *"That list of [communication strategies in LWT] gives you an idea of what options you could have available"* (Client 15) and *"It took the focus of the hearing aid to . . . this is what you can do to make it better for you, so head against a wall in a noisy restaurant, turn the music down, you know you get TV headsets for the television if you're having trouble with Scottish accents"* (Clinician 1). Some client participants noted that the LWT prompted them to identify

additional communication situations to discuss with the audiologist: *"I hadn't thought about the supermarket . . . so it does jog the memory when it needs to"* (Client 8).

Another participant described how the audiologist used information from the LWT to discuss communication strategies which could be used in social situations: *"positioning [myself] to hear and situations where I needed assistance"* (Client 19), while another described how the audiologist used the LWT to discuss hearing aid solutions: *"she was making the point how it [hearing aid] copes in this sort of situation [shown in LWT] and therefore that it [hearing aid] would be beneficial to be in terms of being able to hear better in that sort of scenario"* (Client 16). Finally, participants recognised the benefit of the LWT in keeping records for both the client and audiologist: *"It leaves a permanent record [of what was discussed with audiologist] 'cause you know if you have a meeting with someone you can talk for 15 min about the problems but have no record"* (Client 3).

The LWT could enhance preparedness for the client and clinician. One of the key benefits outlined by participants was the potential application of the LWT in preparing clients for appointments and even facilitating the help-seeking process. Despite only a small number of clients competing the LWT prior to their first appointment, many reflected that completing the LWT pre-appointment would have been valuable to the process: *"I probably would've been more informed [if I had completed the LWT prior]"* (Client 1). One participant reflected *"What the program [LWT] said to me was, this is like pre-empting your visit. The technician's [audiologist] got some information about you"* (Client 4), while others said *"I probably I came [to the appointment] prepared to be saying 'This is the problem I'm having'"* (Client 16). Others saw how the LWT could be used to help clients identify their own hearing difficulties, and even the potential use of the LWT as a communication screener: *"You could screen people using the device [LWT] rather than taking history and everything when you actually saw them . . . and those that have a problem need to come and those who don't have a problem may not need to come . . . or may not need to come so urgently"* (Client 15). Interestingly, client participants also saw the benefits of the LWT in preparing audiologists for appointments, for example: *"They [audiologist] might be able to pre-empt the problem . . . before they see the patient . . . and save them [audiologist] time from asking all the questions . . . which would make the appointment more efficient"* (Client 15).

The LWT improves appointment efficiency. Despite clinicians expressing concern that, prior to using the LWT, it would take too much time to complete, they reflected after the study that *"it doesn't [take more time], because you would be doing COSI [Client Oriented Scale of Improvement] goals anyway"* (Clinician 2) and in fact, *"it [LWT] kept me on track . . . rather than 20 min trying to get to the point [of the client's visit]"* (Clinician 1). Client's also noted that *"it was really just a few minutes at the start of the appointment"* (Client 18) and *"we [client and audiologist] didn't have to go through a long, tedious interview"* (Client 8). Clients noted that the LWT could also be a helpful tool in improving session efficiency for certain clients: *"I think it [LWT] could save a lot more time with people who were maybe were a little bit reticent"* (Client 11).

The LWT aids communication processes. Client participants highlighted how the LWT *"forced the clinician to ask questions and listen to answers"* (Client 7). One client noted how using the LWT *"probably merged a bit into a conversation . . . rather than sort of being too locked into a series of questions . . . ensuring the content was covered naturally"* (Client 17). The clinician participants agreed that the LWT prompted them to ask additional questions about the client's use of communication strategies: *"It [LWT] actually gave you more information, like what they're actually doing [strategies] . . . that's something that I probably got a lot more information than I usually would"* (Clinician 2).

Theme 2: The LWT Supports Person-Centred Audiological care.

Analysis of the client and clinician interviews identified several ways in which the LWT supported person-centred audiological care, including facilitating the client-clinician relationship, making the discussion of sensitive matters easier, helping clients identify the biopsychosocial impacts of their hearing disability, making clients the priority, and normalising hearing experiences to help clients identify strategies.

The LWT facilitates the client-clinician relationship. Both client and clinician participants agreed that the LWT played a role in building the client-clinician relationship: *"I think it [LWT] actually engaged the clinician and the patient, or client more closely"* (Client 7) and *"I think it [LWT] was creating more of a rapport . . . and just took away from the 'I'm the audiologist and you're the client"* (Clinician 1). One participant even reflected that the tool enabled the relationship to begin even before the first appointment: *"I felt like somebody was talking to me already . . . the bond that existed before you even walked in here, before you even met"* (Client 5).

The LWT makes discussing sensitive matters easier. In the context of audiological rehabilitation, the LWT was described by the participants as an effective means of sensitively communicating the diagnosis of a hearing loss: *"It [LWT] obviates the necessity of other people pointing out to you that you've got hearing problems . . . a computer is nicely non-personal . . . at least when you do this sort of thing, you have to come to terms with it yourself"* (Client 13). The LWT was also thought to be helpful in sensitively raising communication difficulties that clients might feel uncomfortable about: *"I can understand how somebody who was a little bit hesitant might find it really really good"* (Client 11), and even help facilitate more thoughtful discussions with family members about difficult communication situations: *"It [LWT] helps them [family members] understand the person with the problem . . . they [family members] might try to assist in certain ways and not be so critical when people didn't pick up what they were saying"* (Client 15).

The LWT can help clients identify the biopsychosocial impacts of their hearing disability. Both clients and clinicians found the LWT particularly helpful in prompting clients to think about the broad ranging impacts of their hearing loss, and the everyday difficulties they encounter as *"they [clients] do tend to say 'I don't think I have any trouble'"* (Clinician 1), with one client even noting that *"It's good to see that there's different ways of looking at it [hearing assessment] now [other than solely instrumental assessment]"* (Client 20). In particular, the pictures of various communication situations prompted clients to think: *"Oh yeah, I have that problem"* (Client 16) and were helpful in making *"the client more aware of the different ranges of hearing situations and how they are currently effective or non-effective"* (Client 7). Importantly, the LWT was also effective in helping clients identify their communication strengths: *"There are many situations in which there are no problems . . . made me aware that I'm competent in a lot of ways"* (Client 20). This focus on communication strengths was noted to be a particularly important given that clients may not always *"think of that [positive] perspective"* (Client 15).

Clients are the priority. The LWT was identified by participants as a method of ensuring the clinician stayed focused on the needs of the client: *"I think it [LWT] was creating more of a rapport so that they knew I was focusing on what their needs were . . . which in turn got me to focus on 'Ok, this is what you want solved', as opposed to me presuming what they were there for."* (Clinician 1). The LWT was also identified as a means of putting the client in the driver's seat: *"They [the client] felt more in control of giving me the information rather than often I'm like, 'Oh, so how do you find the phone?' So more control, they have more control I think"* (Clinician 2).

Relatability of photographs can normalise hearing experiences and help clients identify strategies. Throughout the interviews, clients discussed the relatability of the photographs used in the LWT, and the effectiveness of the photographs in identifying the everyday communication difficulties resulting from hearing loss: *"I think it's [LWT] effective in that it does relate the conditions you're having evaluated to your real life situations"* (Client 21). Further, participants described how the photographs served to reassure clients that other people also experienced communication difficulties: *"By looking at the pictures and realising that there's a lot of other people in those kinds of situations . . . that you're not alone kind of thing"* (Client 10).

Despite these positive reports about the relatability of the photographs used in the LWT, it is important to note that not all clients could relate to the communication situations depicted in the photos: *"The only one that remotely identified with me was the crowd scene or the*

*party scene"* (Client 12). Further, some participants noted that the photographs portrayed people of a homogenous ethnicity and age: *"I think they were all Caucasian people"* (Client 8).

Theme 3: The use of the LWT should be Individualised.

Consistent with the principles of person-centred care, there was wide acknowledgement amongst participants that the use of the LWT should be individualised according to client needs because there was variability in how the LWT was used. The results also highlighted that audiologists provided valuable guidance when using the LWT and that some users value in-person services.

The LWT is used in Varied Ways. Although the online LWT was initially designed to be used by the client at home prior to appointments, and then discussed within the appointment in the context of the client's hearing test results, analysis indicated that the LWT was used by clients and audiologists in varied ways. Despite most clients being sent an electronic link to the online LWT, *"most of my clients didn't do it [LWT] at home"* (Clinician 1) and *"I don't think I had any [LWTs] completed in advance"* (Clinician 2). One of the audiologists gave her clients the choice of how they would like to complete it, noting that *"When I make the appointment I say to them 'I can either send it to you and you can do it online or we can do it together when you come in'"* (Clinician 1). The most common method of completion of the LWT, therefore, was within the audiology appointment time. Although both clinicians indicated a preference for using the LWT at the beginning of appointments *"as part of the case history"* (Clinician 2), clients reflected that *"it doesn't seem to matter which part of the appointment you have it in . . . it wouldn't matter if it was at the start or the end really"* (Client 9).

In regard to ongoing use of the LWT, most clients reported that following completion of the LWT, the audiologists did not commonly refer back to the LWT: *"I don't actually recall us sort of referring back to that [LWT] per se".* Interestingly, a number of clients noted that they saw potential in using the LWT again at a future point in time: *"I know how to access it so if there's a need for me to progress if for whatever reason, if there's any further change to my situation, well . . . there is somewhere I can go to, there is somewhere I can source this information"* (Client 5) and *"I guess you could compare those [LWT responses] overtime"* (Client 17).

Family Member Involvement was Varied. According to participant reports, family members played a limited role in the use of the LWT, most often due to low attendance of family members at audiology appointments: *"I didn't have too many where they actually came along, which is disappointing"* (Clinician 2). When family members did attend appointments where the LWT was used, one family member reported *"I kept as quiet as I could possibly be [during the LWT discussion] . . . I didn't say very much at all"* (Family Member 4), indicating that they were not actively involved in the process. One of the clinicians did note however that she thought the LWT was helpful in provoking discussions about third-party disability. Interestingly, one of the audiologists reflected that the LWT could be used to manage dominant family members: *"It would be interesting to do more of the tool with the significant other to see if it stops them from being the dominant one [and bring the focus] back to the client"* (Clinician 1).

Audiologists Provide Valuable Guidance when Using the LWT. Clients reported that their audiologist provided valuable guidance when using the LWT, with audiologists talking through the instructions with clients, describing the pictures, defining the communication situations, typing in client responses, and discussing these responses throughout the process: *"She [audiologist] spoke to each question"* (Client 2) and *"asked how I coped in the various situations"* (Client 12). The level of input provided by audiologists varied between clients, with audiologists acknowledging that *"some people just needed more . . . some guidance"* and *"it* seems to work whichever way you do it" (Clinician 2). As such, while some clients reported that the audiologist took the lead and completed the tool on behalf of the client by asking them questions: *"I didn't enter anything"* (Client 8), *"I just gave answers [to the audiologist's questions]"* (Client 13), others reported that *"she [audiologist] turned the screen around and gave the mouse to me so I was actually writing it . . . I pretty much went through it [LWT] on my own . . . I think she was more there as a support if I needed it"* (Client 20). When

guidance was provided, this most often included an explanation of various communication strategies such as asking people for clarification, and at the end of appointments linking the results of the hearing test to the LWT responses: *"She [audiologist] did make connections between the situations that were identified on the website and the results of the tests"* (Client 21). Clients noted a preference for the audiologist to enter client responses into the LWT: *"It was probably easier if she [audiologist] uses it. She could put the information in faster than someone who is at home"* (Client 15), however a number of clients also noted that while they were completing the LWT with the audiologist, they could not see the computer screen (Client 13). Overall, however, the majority of clients saw value in going through the tool with the audiologist: *"it would be a bit helpful if I wanted to talk through issues with someone . . . if people are using it by themselves for a diagnosis then you might be entering difficult territory"* (Client 17).

Some LWT Users Value In-person Services. Despite the acknowledged benefits of having access to an online LWT, some clients acknowledged that they preferred to use the tool with their audiologist in an in-person appointment. Client preferences for in-person services centred around their comfort with using technology for healthcare: *"I'm not one of those people who likes talking to screens"* (Client 12), as well as preferring to discuss the results directly with their audiologist: *"It's probably more personalised to do it [LWT] actually with the audiologist rather than do it and give it to the receptionist and then never see it again, and don't know whether anybody's even looked at it"* (Client 8).

Theme 4: Users value Comprehensiveness.

Analysis of the interviews revealed a common finding that the users of the LWT valued comprehensiveness in a tool such as the LWT, and as such, there was potential for the LWT to be further developed to enhance its application in audiological rehabilitation.

The LWT is Comprehensive. Overall, participants agreed that the LWT was a comprehensive tool which offered *"plenty of options"* (Client 11), along with *"a list of different perspectives"* (Client 15).

There is Potential for the LWT to Ask More. Participants did acknowledge however that the LWT had the potential to ask more information and to give more information to clients with hearing loss. Participants suggested that the tool could ask for more specific information to give *"the audiologist a bit of a better indication of what your problems are"* (Client 3), especially *"if you had a problem and you felt you wanted to do a few different things"* (Client 5). Participants also agreed that the LWT could be more comprehensive by asking clients to identify more potential communication strategies.

There is Potential for the LWT to Give More. Participants also noted that there was potential for the LWT to include more communication scenarios, more specific communication strategies, giving more *"examples of what you can hear and what you can't hear"* (Client 3), and providing more background information about the communication scenarios *"with maybe even more of a breakdown of the different scenarios rather than just the pictures . . . more clarification of the pictures"* (Client 10).

Theme 5: Users Value Accessibility of the LWT.

Participants identified the importance of being able to readily access tools such using technology such as the LWT. Participants thus emphasised the importance of having a tool which is easy to use and is aesthetically pleasing, especially given the potential challenges of using technology.

Usability of the LWT is Valued. It was apparent from the interviews that the usability of the LWT is an important consideration in order to maximise accessibility for users. The majority of participants noted that the LWT *"was very straight forward"* (Client 8) and *"pretty easy to understand"* (Client 9), especially given its logical step-by-step approach. One of the clinicians noted however that *"it's not so intuitive"* (Clinician 2) and may be difficult for some clients to use. This was especially true for clients with other co-occurring conditions such as vision loss, which could affect their access to the LWT, and therefore has implications for the design of the tool, including font size: *"You might make the text [of the LWT] bigger for people with poor vision"* (Client 15).

The Aesthetics of the LWT Enhances its Accessibility. Overwhelmingly, the participants agreed that the LWT was "well presented" (Client 17) with its design and aesthetics positively enhanced it's accessibility: *"When you're faced with a website that you don't have to think about . . . it means it's [the LWT] got to be good"* (Client 11). Participants also reported that the *"pictures were effective in picking up [communication] situations"* (Client 21). Some participants did note however that *"the photos were a bit busy . . . and quite small"* (Client 17).

Technology can be Challenging. Given the online format of the LWT, participants did make reference to the inherent challenges of an online platform, especially for *"older, much older people [who] may have a little bit of difficulty with it . . . if you don't understand the computer"* (Client 10). Some participants even expressed a preference for the LWT to be also provided in a *"printable format"* (Client 2) so they could *"fill it out"* (Client 15) in paper form, especially given some clients reported some technical difficulties with sending the electronic copy of the completed LWT to their clinician. Other participants with limited computer experience noted however that despite this lack of technological experience, the LWT was easy to use.

## 4. Discussion

The present study investigated the experiences of clients with hearing loss and their audiologists in using an online audiological rehabilitation tool, the LWT, and specifically, the extent to which the tool facilitated person-centred audiological care. We aimed to explore the experiences of clients and their audiologists in using the tool and determine the extent to which the LWT facilitates the provision of person-centred care. In doing so, we explored the effectiveness of Living Well in focusing audiologists' talk on communication in everyday life rather than just hearing aids when working with adults with hearing loss and their families. Results of this study produced two overarching ideas. The first of these was that Living Well embodied person-centred care, as indicated by themes 1–3; the second was that users value the website's functionality, shown in themes 4 and 5.

The Living Well tool embodied person-centred care as indicated by the themes LWT supports person-centred audiological care, LWT enhances audiological care, and Use of LWT should be individualised. Participants in the present study showed the Living Well tool is adaptable to individuals' heterogeneity by describing the variability in tool operator (client, clinician or together), how the tool is used (providing a reference point for the rest of the session, informing session focus and directionality) and how family members were or could be involved. Previous research has identified the importance of taking a person-centred approach to healthcare [4]; this was shown in the present study by helping clients to identify their strengths, weaknesses and needs of communication via relatable photographs. Taking a person-centred perspective allows for recognition of the patient as a complete person [4], which is related to increased knowledge, improved service experience, and better outcomes [21,22]. Healthcare providers taking an interest in non-clinical concerns can help foster emotional wellbeing, which is predictive of physical health [23].

Participants described how the LWT facilitated communication, facilitating the relationship between clients and clinicians, and easing the discussion of sensitive matters. This description is consistent with [4] characterisation of person-centred care being facilitated by enablers including clinician-client communication, access to care, and a team-based approach. They are also consistent with previous findings that show person-centred communication such as showing sensitivity to emotional concerns fosters the therapeutic alliance [24]. This person-centred communication in turn can lead to better outcomes [25]. Additionally, person-centred communication has further been linked to improved treatment compliance [26], and has been suggested as an approach to improve patient dignity and reduce stigma [27].

In the present study, clients and clinicians alike commented on the ways in which the LWT integrated with existing models and activities of care without diminishing traditional practice. Both client and clinician participants reported that the LWT improved preparedness for audiological rehabilitation, and therefore promoted greater session effi-

ciency. Importantly, participants also reported that the LWT did not add any more time to appointments. Identifying these benefits is important given clinicians report time as a barrier to implementing person-centred care [28–30]. Indeed, recent research has found that PFCC can be flexibly applied and produces a more collegiate environment for clinicians that is perceived by patients [31].

In addition to enhancing the information that is received from clients, the LWT also has capacity to help clients gain additional insight; the LWT helped clients identify the biopsychosocial impact of their hearing disability. Many clients underestimate the impact of their hearing loss until rehabilitation has commenced. The graduated nature of most cases of hearing loss means many clients do not realise what they are increasingly missing out on [32] and the psychosocial effects this can have [27,33]. The provision of biopsychosocial feedback via an inexpensive, accessible, and non-invasive tool such as Living Well has the potential to improve insight for clients, such as their communication limitations and needs. More importantly, clients in the present study reported that the LWT helped them to recognise the communication strengths they already possessed, which is aligned with the strengths-based approach used in person-centred care [15].

In recognising client strengths and providing individualised care, information and its presentation can be tailored to suit the user [34], which can increase engagement with technology. Personal relevance plays a role in perceived usefulness [35], which is a strong influence on older users' adoption of technology [17,36]. The desire for tailored information has been reflected in the present study, which highlighted the importance of platform functionality and practicality via the third and fourth themes, Users value comprehensiveness and Users value accessibility of LWT respectively. In terms of comprehensiveness, participants in this study felt the tool could have both given more and asked more of them, suggesting that they valued the provision of information and the validation of having their experiences heard. Substantive informational exchange is advisable for internet-based hearing rehabilitation. A recent content analysis of interviews with people who received an internet-based intervention for hearing loss found that the program increased participants' self-esteem by increasing their hearing loss knowledge and their self-efficacy for using that knowledge [37].

Accessibility of such programs has been commented on extensively in the literature, which has found that designing for older users requires careful consideration of layout, use of blank space and proximity and choice of colours [38,39], and a range of measures and checklists have been collated to guide the development of technology aimed at assisting adults [40]. Research by Lee and Coughlin [41] identified ten factors that facilitate older adults' use of technology; these are value, usability, affordability, accessibility, technical support, social support, emotion, independence, experience, and confidence. Several of these were mentioned by this study's participants, who had commented on their own confidence in using the site and its usability for individuals with additional impairments. Our findings of the LWT's usability contribute to broader findings for the validity of using visual aids in exploring needs; recent research has used a photovoice methodology where participants documented their hearing and communication experiences by taking photographs, which produced highly person-centred results. These included tailored counselling and recommendations, improved insight into lifestyles and interactions between communication partners, and improved relationships with clinicians [42].

In terms of clinical implications, the Living Well tool provides a range of options for clinicians in future. As shown in the present research, it can be administered within the clinic with the help of the clinician, or can be used at home, either alone or with a family member. Indeed, input from family members is strongly encouraged; research has shown family-centred care improves communication within families and reduces third-party disability [43,44]. Family inclusion can provide a comprehensive overview of family communication difficulties [45], and increases the likelihood of strong family member support if their needs are heard and met. Currently, the LWT targets individuals and limits family inclusion to the role of helping the client with a set goal. The present results

indicate support for expanding this platform to focus on families. Ideally, this focus would move beyond family involvement, whereby family members are asked about difficulties encountered by the person with hearing loss, and move towards family-centredness, where discussion is centred on the communication goals, needs and preferences of individuals within the family and the family as a whole.

The present study contained some limitations. Firstly, we had limited control over which participants were invited. Clients of the participating clinics were invited directly by the audiologist; accordingly, we have no knowledge of which clients were not invited or for what reasons. Secondly, many of the interviews were conducted in the clinic of the treating audiologist, which may have influenced the interview data. Despite assurances of confidentiality, clients may have been concerned their responses could be overheard by or provided to clinicians. We conducted interviews in the clinic to maximise response rates; however, future research may be able to circumvent potential influence on interview responses by conducting all interviews in a separate setting. Finally, we aimed to gain the perspectives of family members, yet only two clients brought family members to their appointments. Although low levels of family attendance in audiology are common [45], increased family member attendance will facilitate the increased implementation of family-centred care, rather than relying on asking clients for their perceptions of the family members' needs and goals.

Despite these limitations, the present study was the first of its kind in examining client and clinician experiences of using this LWT, and thus has provided important insights into how much an online tool can be used to enhance audiological care. The LWT tool provided participants with a platform to discuss their hearing and communication needs with their audiologist, in a way that has promoted person-centred care via individualised, comprehensive, accessible care that is suitable for an older audience. The use of the LWT can be expanded to provide family-centred care by including the needs and perspectives of family members.

**Author Contributions:** Conceptualization, N.S., K.T., C.M., K.E. and C.L., Methodology, N.S., K.T., C.M., K.E. and C.L., Formal Analysis, N.S., K.T., Investigation, N.S., K.T., K.E., Data Curation, N.S., K.T., K.E., C.M.; Writing—Original Draft Preparation, N.S. and K.T.; Writing—Review and Editing, N.S., K.T., C.M., K.E. and C.L.; Project Administration, N.S. and K.T., Funding Acquisition, N.S., K.T., C.M., K.E. and C.L. All authors have read and agreed to the published version of the manuscript.

**Funding:** Components of this research have been presented at Hearing Across the Lifespan; Cernobbio, Italy; 7 June 2018. This research was funded by the Ida Institute Research Committee Granting Scheme.

**Institutional Review Board Statement:** Ethical clearance for this study was obtained from The University of Queensland Human Research Ethics Committee (Approval # 2016001740).

**Informed Consent Statement:** Informed consent was obtained from all participants involved in the study.

**Data Availability Statement:** Not applicable.

**Acknowledgments:** The authors would like to thank all participants for their invaluable contribution to the study. We would also like to thank the participating clinicians for their assistance with participant recruitment.

**Conflicts of Interest:** The authors declare no conflict of interest.

## Appendix A

### Client Interview Guide

1. Can you tell me about your experience of using the Living Well tool?

➢ How did it help you in identifying your hearing and communication needs?
➢ How did it help you in identifying your communication strengths?

➢ How did it help you in identifying strategies you could use to improve your communication?

➢ How easy was it to use?

2. Once you completed the first part of the Living Well tool, how did the audiologist use it with you in your appointment?

➢ What did you talk about?

➢ How did you feel about the experience?

➢ Tell me about the positive things it brought to the appointment?

➢ Tell me about how it could have been used differently?

3. What do you see as the strengths of identifying your hearing and communication needs in this way?

➢ From your perspective? From your significant other's perspective? From the audiologist's perspective?

4. What do you see as the challenges of identifying and managing your hearing and communication needs in this way?

➢ From your perspective? From your significant other's perspective? From the audiologist's perspective?

5. How was this appointment using the Living well tool different to other appointments you have had? What was different?

6. If you could change something about the tool, what would you change?

**Audiologist Interview guide**

1. Can you tell me about your experience of using the Living Well tool?

➢ How did it help you in identifying the hearing and communication needs of the client with hearing loss?

➢ How did it help you in identifying the hearing and communication needs of the client's significant other?

➢ How did it help in identifying the client and significant other's communication strengths?

➢ How did it help you in identifying strategies for the client and significant other to improve their communication?

➢ How easy was it to use?

2. How did you use the tool in the appointment?

➢ What did you talk about?

1. How did you feel about the experience?

➢ How did you involve the significant other in the discussion?

➢ Tell me about the positive things it brought to the appointment?

➢ Tell me about how it could have been used differently?

3. What do you see as the strengths of identifying client's hearing and communication needs in this way?

➢ From the client with hearing loss's perspective? From the significant other's perspective? From your perspective?

4. What do you see as the challenges of identifying and managing your hearing and communication needs in this way?

➢ From the client with hearing loss's perspective? From the significant other's perspective? From your perspective?

5. How was this appointment using the Living well tool different to typical appointments you have? What was different?

6. If you could change something about the tool, what would you change?

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
