# Peer review of "Using an Online Tool to Apply a Person-Centred Approach in Audiological Rehabilitation: A Pilot Study"

_audiolres, doi:10.3390/audiolres12060060_

Round 1
Reviewer 1 Report
The authors have successfully demonstrated the person-centered Approach in Audiological Rehabilitation
Reviewer 2 Report
Comments for the author:
Thank you for the opportunity to review this manuscript. This study aimed to “explore the experiences of adult clients with hearing loss and their audiologists in using an online tool, the Living Well Tool (LWT), during initial audiology appointments”. These objectives are of great importance to the field of audiology, both from educational and clinical practice perspectives, and the authors make a fine argument for that. However, I do have some minor concerns regarding this manuscript that need to be clarified.
Abstract:
Here, you are presenting 24 clients who participated in the study. On page 3; line 109, 24 participants were recruited, but, on page 5; line 163, 23 participants were interviewed. In what sense did the one person (1 person) included in those who participated in the study (24) but not interviewed (23) participated in the study?
Background:
The background is overall well written, nevertheless, I would like to make some suggestions that may strengthen the manuscript. Para 2, page 2, lines 59-75: this section may benefit from recent research in eHealth evaluated in clinical settings by Dr. Ferguson/Dr. Maidment/Dr. Malmberg. Furthermore, para 4, page 2, lines 85-99: this section needs some clarification regarding the importance of being “prepared” before a first audiology appointment, and the importance of a person with HL being able and knowledgeable in how to identify individual goals and needs within AR. I highly agree with the authors that this is relevant and central to reaching beneficial outcomes of AR but there is a lack of research support for that in this section.
Material and methods:
The material and methods section is overall lacking clarity on ‘how’ the participants were recruited for this study. The reader gets the information on ‘where’ and which inclusion criteria that yielded, but not ‘how’. This should be clarified. For example, one of the criteria to participate was (b) self-reported hearing loss; how was this ensured during the recruiting process?
Also, 12 of the 24 participants currently wore hearing aids, and I assume that this was not their first audiology appointment. The background section highlights the importance of presenting LWT prior to the participants’ first audiology appointment to identify when and where it is most important for them to communicate. My point here is that many hearing aid users may have already identified communication failures and are more experienced in living well with hearing loss compared to persons with self-reported hearing loss that don’t use hearing aids. Could this be something affecting your results? Also, that is something that should be discussed in the discussion section.
Page 3, lines 120-123: I’m not sure that I’m getting this right. Are the appointment outcomes (n=25) related to each participant (n=24) or did one of the participants had combined appointment outcomes? (or actually, 2, that is if 23 clients were interviewed)
Page 3, line 124. Clients should be in bolded text.
Page 5, line 166: you should state the average time for the interviews and where you held the interviews.
Results
The results section is quite long, and I suggest removing Tables 1 to 5 since the information in those is quite repetitive. Instead, a table presenting themes and sub-themes could be added.
Page 6, line 218: “The LWT could enhance preparedness….” In Table 1 this heading is written without the word ‘could’. This should be the same.
Page 7, line 270: change to: …the LWT was described by the participants as an effective….
Page 7, lines 292-293: Something is missing in this sentence. Given what?
Page 8, lines 330-331: This is not how you presented this information in the method section. In the method section, not all clients were sent an electronic link. It was sent “in most cases”. This should be clarified.
Page 8, lines 334-335: This should be presented in the method section as a part of the study procedure.
Page 9, line 367: the quote starting with “it seems to work…” should be in italics.
Page 10, lines 396-411: This section is not divided into subthemes like the other four themes. This should be consistent.
Discussion
The discussion is well written, although I have a few suggestions for improvement:
Page 12, line 468: remove the colon after ..2014b)
Page 12, line 474: Add (2014) between colleagues and characterisation.
Page 12, line 478: Clarify what you are referring to with “This in turn….”.
Page 13, line 562: What is MPOC-A?
Page 13, line 561-564: The sentence “Similarly, many participants …..been more appropriate.” I’m not sure I’m following this correctly, this information is new. Is MPOC-A something the current study participants answered? In that case, it should be presented in the methods section as a part of the current study method.
Page 13, lines 564-565: The sentence “The ‘not-applicable’ response…research.” What makes a patient familiar with person-centred care? This is an interesting statement but I think it needs some more explanation.
